# Patient Perspectives on Portal-Based Anxiety and Depression Screening in HIV Care: A Qualitative Study Using the Consolidated Framework for Implementation Research

**DOI:** 10.3390/ijerph21060692

**Published:** 2024-05-28

**Authors:** Jacob A. Walker, Erin M. Staab, Jessica P. Ridgway, Jessica Schmitt, Melissa I. Franco, Scott Hunter, Darnell Motley, Neda Laiteerapong

**Affiliations:** 1Department of Medicine, University of Colorado School of Medicine, Aurora, CO 80045, USA; 2Department of Medicine, University of Chicago, Chicago, IL 60637, USAnlaiteer@bsd.uchicago.edu (N.L.); 3Chicago Center for HIV Elimination, Department of Medicine, University of Chicago, Chicago, IL 60637, USA; 4WCG Clinical Endpoint Solutions, Hamilton, NJ 08540, USA

**Keywords:** anxiety, depression, HIV, patient portals, patient acceptance of health care

## Abstract

Electronic patient portals represent a promising means of integrating mental health assessments into HIV care where anxiety and depression are highly prevalent. Patient attitudes toward portal-based mental health screening within HIV clinics have not been well described. The aim of this formative qualitative study is to characterize the patient-perceived facilitators and barriers to portal-based anxiety and depression screening within HIV care in order to inform implementation strategies for mental health screening. Twelve adult HIV clinic patients participated in semi-structured interviews that were audio recorded and transcribed. The transcripts were coded using constructs from the Consolidated Framework for Implementation Research and analyzed thematically to identify the barriers to and facilitators of portal-based anxiety and depression screening. Facilitators included an absence of alternative screening methods, an approachable design, perceived adaptability, high compatibility with HIV care, the potential for linkage to treatment, an increased self-awareness of mental health conditions, the ability to bundle screening with clinic visits, and communicating an action plan for results. The barriers included difficulty navigating the patient portal system, a lack of technical support, stigmatization from the healthcare system, care team response times, and the novelty of using patient portals for communication. The patients in the HIV clinic viewed the use of a portal-based anxiety and depression screening tool as highly compatible with routine HIV care. Technical difficulties, follow-up concerns, and a fear of stigmatization were commonly perceived as barriers to portal use. The results of this study can be used to inform implementation strategies when designing or incorporating portal-based mental health screening into other HIV care settings.

## 1. Introduction

Anxiety and depression are highly prevalent among people living with HIV (PWH), with the estimated prevalence of anxiety ranging from 13 to 80% [1,2,3] and depression ranging from 14 to 79% worldwide [1,2,4]. The high rate of mental health conditions in PWH is hypothesized to be driven by overlapping sociodemographic vulnerability, medical comorbidities, and HIV-associated stigma [1]. PWH with comorbid anxiety and depression have lower rates of engagement with HIV care and lower antiretroviral therapy adherence [4,5,6,7,8,9,10]. Despite the high prevalence of anxiety and depression among PWH, these conditions are often underdiagnosed and undertreated, with as many as half of all cases going unrecognized by clinicians [11].

Although the need to screen for and treat anxiety and depression in PWH is great, there are many barriers to implementing systematic mental health screening in HIV care settings, including a high clinical workload, insufficient time, a lack of training in mental health, and inadequate referral resources [12,13,14]. Strategies to overcome these barriers include the use of designated ancillary staff to perform mental health assessments, electronic anxiety and depression screening during clinic visits, and the co-location of HIV and mental health services [15,16,17]. However, these strategies require additional time and staff, which are limited in many HIV care programs [18].

A promising new strategy for integrating mental health assessments into HIV care is the use of electronic patient portals. Patient portals are encrypted websites that give patients access to their health information via a web connection and allow for secure messaging in compliance with the United States (U.S.) Health Insurance Portability and Accountability Act [19]. Portals can be created such that patients can view results, schedule or cancel appointments, request medication refills, and communicate with their providers [20]. Patient portals that are linked, or tethered, to an electronic health record (EHR) enable data flow between the healthcare system and patients. As a result of the Centers for Medicare & Medicaid Services EHR incentive program, 87% of ambulatory care practices in the U.S. have EHRs, and many also have patient portals [21,22].

Patient portals are also a potential avenue for improving outcomes and quality of care in both HIV care and mental health care [23]. Internet-based technologies, such as email communication with healthcare providers and health-based smartphone apps, have already been shown to be effective for improving health outcomes among PWH [24,25]. Patients who use portals have fewer no-show appointments, higher levels of care satisfaction, and greater medication adherence rates [26,27,28,29,30]. Portals may also help to achieve greater health equity via accessibility, as patients with disabilities, chronic conditions, and frequent health care utilization are more likely to engage with portals [31,32]. With the increasing use of technology in daily life, the majority of PWH are interested in using electronic patient portals for communicating with their providers [33].

However, there is little published research on how patients perceive the use of patient portals for mental health screening within HIV care settings. Portal-based depression screening has been met with skepticism from patients in other specialty care settings, noting concerns with privacy, data use, and the stigmatization of mental health care [34,35,36]. It is unknown how PWH may perceive similar electronic screening strategies. The aim of this formative qualitative study is to characterize the patient-perceived facilitators and barriers to portal-based anxiety and depression screening within an HIV care setting using the Consolidated Framework for Implementation Research (CFIR) in order to inform the implementation of strategies for mental health screening. CFIR is a widely adapted implementation framework used to identify the facilitators and barriers to interventions applied at the local level [37,38]. The use of this framework to guide semi-structured patient interviews enables the tailoring of implementation strategies to meet local HIV clinic needs while allowing for a comparison with the larger body of work on depression and anxiety screening in subspecialty care.

## 2. Materials and Methods

### 2.1. Study Design

This study utilized qualitative methods to inform the design and implementation of an electronic patient portal-based strategy for the screening and symptom monitoring of anxiety and depression within an HIV care clinic. A phenomenological approach was utilized for the stakeholder input analysis. Potential facilitators and barriers to uptake were identified via a thematic analysis of the semi-structured interviews undertaken with the patient and clinician stakeholders, based on CFIR constructs. The results of the clinician interviews have been published separately [39]. Verbal informed consent was obtained from all subjects involved in the study. The study design and verbal consent script were reviewed and approved by the University of Chicago Institutional Review Board (IRB20-1313).

### 2.2. Study Setting

This study took place at the University of Chicago Ryan White HIV Care Clinic, located in the Hyde Park neighborhood on the South Side of Chicago. This large multidisciplinary clinic provides comprehensive care for adults with HIV, including the screening and management of mental health needs. The health system’s EHR is provided by Epic, which uses the MyChart electronic patient portal for patient communication, medical record access, prescription management, and appointment scheduling. An electronic patient portal-based system for anxiety and depression screening for use in the University of Chicago’s adult primary care clinic was under evaluation concurrent to this study [40,41].

### 2.3. Study Sample

Eligible participants included all English-speaking adults (≥18 year of age) who received HIV care at the University of Chicago Ryan White HIV Care Clinic. A purposive sampling method was conducted to target a 1:1 recruitment ratio of PWH with anxiety/depression and PWH without anxiety/depression. The research team (JS) reached out to individual HIV providers for permission to contact potentially eligible participants. An HIV social worker (JS) then called potentially eligible participants to introduce the goals of the study and forwarded contact information to the project manager if there was interest. The project manager (MF) then reviewed the medical record for eligibility and contacted participants to obtain consent and schedule an interview. Individuals were excluded from the study if they had a documented serious mental illness, such as a personality disorder, schizophrenia, psychotic disorder, autism spectrum disorder, or other neurodevelopmental disorders. This exclusion criteria were chosen as patients with pre-existing serious mental illness are not the main population of interest for anxiety and depression screening. Individuals were also excluded if enrolled in hospice care. The team members involved in obtaining consent, scheduling interviews, conducting interviews, and analyzing transcripts did not have pre-existing relationships with the participants.

### 2.4. Data Collection

Patient demographic information was collected prior to interviews via a brief online survey. Audio-only interviews were conducted via Zoom videoconferencing using a semi-structured format. Patient interviews were conducted by a male doctorate-level clinical health psychologist using a predetermined interview protocol. The protocol was iteratively developed by members of the study team with questions informed by the CFIR Interview Guide Tool [42]. The protocol included general questions about anxiety and depression screening and management within the HIV clinic and framed the relevant intervention as the use of an electronic portal for anxiety and depression screening and management. The protocol was not pilot-tested with patients prior to use. Participants were introduced to the goals of the study and the background of the research team at the start of each interview. Videoconferencing was utilized to demonstrate a sample of the portal-based screening tool to participants during the interview. The samples included screenshots of an introduction page and a common depression screening tool, the Patient Health Questionnaire-9, as they would appear in the patient portal. Samples were gathered from the portal-based screening tool being studied concurrently in the University of Chicago adult primary care clinics [40,41]. The interviews allowed up to 90 min for completion and usually lasted between 30 to 60 min. There were no repeat interviews. Interviews were conducted from May 2021 to July 2021. Interviews were audio-recorded, transcribed, and de-identified prior to analysis. Written records and video recordings were not kept during the interviews to reduce the risk of privacy loss. The transcripts were not reviewed by the participants and the participants were not asked to provide feedback on the findings prior to publication. Patients were aware in advance that they would receive USD 40 for completing an interview. A copy of the interview guide is included in the electronic Appendix A.

### 2.5. Data Analysis

A thematic analysis of the interview transcripts was performed using a deductive approach [43]. An initial codebook was created using the CFIR Codebook as a template [44]. Transcript coding was grouped according to the five CFIR domains: (1) Intervention characteristics, (2) Outer setting, (3) Inner setting, (4) Characteristics of individuals, and (5) Process. Although the interview questions were designed to address specific CFIR domains, the responses to any portion of the interview were coded to all relevant CFIR domains. During an initial coding phase, the research members (JW, ES, NL) collaboratively coded transcripts and refined the CFIR codebook as needed to capture emerging themes and subthemes. Once a coding framework was established, all remaining transcripts were reviewed by two independent coders. The coders met upon the completion of each transcript to review the coding consistency and reach a consensus. Theme saturation was determined a priori as reaching two interviews with no new codes [45]. An analysis of the consensus coding was conducted using Dedoose, a web-based data analysis software, www.dedoose.com (accessed on 20 October 2021).

## 3. Results

### 3.1. Participant Characteristics

The study team initially set out to conduct focus groups, but many participants declined to participate or did not arrive at the scheduled interviews. Of the 59 adults approached for recruitment, 47 declined to participate. The reasons for non-participation were not recorded. As a result, a total of 12 patients participated in 7 interviews ranging in size from 1 to 3 participants. The participants ranged in age from 26 to 57 years, with a mean age of 39 years (SD 10 years). On average, participants had been living with HIV for 10 years (SD 8). The participants were largely men (83%) and three quarters self-identified as Black/African American (75%). Half of the participants carried a current diagnosis of depression or anxiety in the medical record (50%). Approximately two thirds of the participants reported the current use of the electronic patient portal (67%), with more reporting intent to use the patient portal in the future (17%). The participant characteristics are further summarized in Table 1.

### 3.2. Thematic Analysis

All interviews spanned multiple CFIR domains, with most participants responding to all five domains within a single interview. Figure 1 demonstrates the codes identified during data analysis, distributed by the CFIR domain and construct. The key facilitators and barriers to anxiety and depression screening implementation are presented according to the CFIR domain (Table 2), with representative quotes below.

#### 3.2.1. Intervention Characteristics

This domain includes the attributes of the intervention (the electronic portal-based screening tool) that influence successful implementation. The identified constructs within the intervention characteristics included evidence strength and quality, relative advantage, adaptability, complexity, and design quality and packaging.

##### Facilitator: Absence of Alternative Anxiety and Depression Screening Methods

The only anxiety and depression screening method that patients were already familiar with was direct conversation with their care team. The participants acknowledged that initiating conversations about mental health may be difficult for many patients or may not be within the skill set of their HIV provider. Few expected their HIV provider to initiate anxiety and depression screening conversations de novo, citing a patient’s responsibility to report symptoms. In this context, the portal was seen as an easy and novel method to initiate conversations around anxiety and depression that may otherwise never occur.


*“I hid my anxiety for a long time. I was ashamed, I was afraid of talking about it. I was dealing on my own and I think somebody actually in my circle, my friends circle told me that I should look for help. And it was not easy to open up to a doctor, so if I were to do it through the chart and know that I’m going to get the right help, it will probably be a lot easier for me to do so”.*
Participant 1 (52 year-old male)

##### Facilitator: Simple and Approachable Portal Design

Patients commented on the portal screening tool’s simple appearance, appropriate length, and anticipated ease of use. None of the patients expressed dissatisfaction with the tool’s design. They also found the brief instructions to be inviting: “They’re trying to make sure that we do the right thing and take our meds properly. It’s caring to me, I like it [Participant 2 (36-year-old male)].” Two participants felt that increasing interactivity with a moveable slider for responses could make the system more appealing.

##### Facilitator: Adaptability of Screening Frequency

Responses varied when asked about the preferred frequency of screening, but patients consistently anticipated that the portal was capable of meeting any frequency need. Several respondents conceptualized the portal less as a screening tool and more as a symptom-tracking system with which they could relay mood and anxiety symptoms to their care team as needed. The ability to adapt the frequency of screening to meet patient needs was seen as a positive aspect of the portal, despite no explicit mention by the interviewers that the portal would have that capability.


*“I think it’s just depending on the person. Because someone might need it more than other people. Somebody else might be more reserved. Someone else might just need to talk to people and might need that assistance”.*
Participant 3 (41-year-old male) on preferred frequency of screening

##### Barrier: The Electronic Patient Portal System Is Difficult for Patients to Navigate

Most described the screening tool itself as simple and inviting, but expressed frustration with the visual noise within the patient portal system (MyChart), including hard-to-read fine print, an excessive number of alerts, and a complex menu system. Having to navigate the patient portal would deter some patients from accessing the screening portal. Even among respondents more comfortable with patient portals, difficulties accessing screening results were anticipated, informed by difficulties accessing lab results within the same system.


*“Yes, it’s a little hostile when you go through [MyChart]. I think you’ve got to play with it a little bit like I did. It’s a lot to go through. Like when it’s first time like downloading the app, and trying to get a handle on everything, looking at your doctors, how to message your doctors, I think that’s the hardest part…”*
Participant 4 (25-year-old female)

#### 3.2.2. Outer Setting

The outer setting domain includes constructs covering factors from outside of the organization that influence implementation. Nearly all coded responses within this domain involved the needs and resources of those served by the organization.

##### Barrier: Fears of Mental Health Stigmatization from the Health Care System

The privacy of the screening results was highly prioritized among some patients. Most respondents voiced a preference that portal screening results only be sent to known members of the care team. Patients preferred that a familiar staff member (usually their HIV provider or social worker) follow-up on results rather than someone they had never met. Patients felt that it would be awkward or inappropriate to discuss their results with medical assistants or front-desk staff, for example. There were also concerns that a positive screening could lead to the automatic documentation of anxiety or depression in the medical record, diagnoses that may be inaccurate. One patient voiced concern that positive screening results in the medical record could also lead to differential treatment from the health care system.


*“…the scheduler, the person taking vitals, confirming your medication, all of that, right? I’m really not warm and fuzzy about that being who notifies you”.*
Participant 5 (34-year-old male)

#### 3.2.3. Inner Setting

The inner setting domain includes constructs relevant to the specific organization’s inner working and preparedness for implementation. The identified constructs included networks and communications, the implementation climate, and readiness for implementation. The subcodes included tension for change, compatibility, available resources, and access to knowledge and information.

##### Facilitator: Anxiety and Depression Screening Is Highly Compatible with Routine HIV Care

The HIV clinic was universally seen as an appropriate venue for anxiety and depression screening. Anxiety and depression screening were viewed as a normal part of medical care, akin to checking vital signs. Some strongly expressed that screening is needed among HIV-positive peers, even expressing that rigorous mental health screening is a matter of patient safety. A few respondents expressed skepticism that their HIV provider would be able to adequately respond to mental health needs, but also confidence that their HIV team would know how to connect them to mental health providers.


*“I look at the mental health care and the ID healthcare as being interlinked in the sense that better mental health care would seem like it would result in better outcomes for an ID patient and vice versa, more of that holistic approach to health and making sure that someone who is an ID clinic and does need access to mental health care is able to get those services”.*
Participant 5 (34-year-old male)

##### Barrier: Lack of Technical Support for Patient Portal Users

Patients expressed frustration with navigating the patient portal system. They recalled being advised by clinicians to sign up for the patient portal but did not routinely receive guidance on how to use it or why it was being recommended. They felt that there were few tutorials on how to use the system once they had signed up.


*“I don’t know what MyChart is really designed for. Somebody said it has my team on it but nobody has ever explained to me, what you could do with MyChart other than the results and your appointments will be on MyChart”.*
Participant 6 (36-year-old male)

##### Barrier: The Portal Is an Extraneous Means of Communication with the Care Team

Portal-based screening was seen as being outside of normal patient–clinician communication methods. Few patients reported already using the patient portal to communicate with their care team. Most preferred the use of phone calls or text messages for reminders to complete screening tasks and for the follow-up of screening results. Phone calls, text messages, and in-person visits were felt to be more “direct” forms of communication than electronic messaging systems such as email or the patient portal; these were thus preferred for important clinical care matters. Three respondents, who were not already using it for other tasks, expressed that the portal was an extraneous addition to their care.


*“I believe onsite, hands-on, in-person talk with my doctor, so I can get the information that I need. [The portal] is a great asset for people who need to use it but, me, I don’t have time for it. I have to be in person”.*
Participant 7 (57-year-old male)

##### Barrier: The Utility of Portal-Based Screening Is Dependent on Care Team Responsiveness

Patients expressed doubt that their care team would be notified of their screening results or respond to the screening results in a timely manner. These expectations were shaped by past experiences with the care team, especially attempting to reach busy HIV providers. The expectations of how long it should take the care team to respond to screening results varied. Responsiveness was seen as a potential safety concern if strongly positive screening results were not tended to quickly, as they could indicate a patient at risk of self-harm. Automated messages or direct replies from staff stating that the results were passed to an appropriate clinician (usually a physician or social worker) were seen as potentially reassuring interventions.


*“…If I type that in like, “Doctor, oh, I’m having anxiety and depression today.” Then, my letter should read high rate because there is a concern here. There is an emergency here. Then, have a quick answer. Have those tools in place, so they can address those issues. If you’re going to use [the portal] have the tools”.*
Participant 7 (57-year-old male)


*“I’ve had the feeling where I needed to talk to my doctor, right away, and I’m not able to get through the phone service. So, I go on MyChart. I usually hear back from them maybe two or three days later and I don’t really think that’s sufficient. I could have been going through really, physical and mental breakdown and I really needed to talk to my doctor have her assure me that everything is going to be okay or come in for an emergency check-in just to ease my mind and see if my anxiety is more or if my depression is more”.*
Participant 4 (25-year-old female)

#### 3.2.4. Characteristics of Individuals

This domain includes constructs pertaining to the behaviors and beliefs of individuals within a network. The identified constructs included knowledge and beliefs about the innovation, individual state of change, individual identification with the organization, and other personal attributes.

##### Facilitator: Anxiety and Depression Screening Is an Important Part of Linkage to Anxiety and Depression Treatment

Screening for anxiety and depression, regardless of modality, was seen as a positive step toward improved mental health. Patients described screening as applying forward momentum to anxiety and depression management through increased diagnosis, the increased awareness of providers regarding symptoms, and an increased connection to treatment. The ability to track screening results over time via the patient portal was seen as a potential asset to treatment.


*“I think that we should be aware of [anxiety and depression]. That it be known that it is an illness and you need to get help if you need it. Don’t be invasive about it. You know what I’m saying? Don’t be aggressive. But I think that it should be talked about. It should be... Do you know what I’m saying? It should be confronted.”*
Participant 8 (38-year-old male)


*“Oh, I would definitely think [screening] would be needed and would be successful and would be an asset to the treatment because when you’re going through... When you’re going through things, it’s a lot. And some people are going through so much. That’s a lot that you deal with here in this world today. And if my doctor asked anything about anxiety or depression, I would definitely be open to that.”*
Participant 8 (38-year-old male)

##### Facilitator: Screening Questions May Increase Self-Awareness of Depression and Anxiety Symptoms

Several respondents with previously diagnosed depression and anxiety suggested that screening questions are a means of increasing the user’s self-awareness of their own depression and anxiety symptoms. Citing a lack of awareness of depressive symptoms in the general public, some patients expressed that the screening questions could serve as a conversation starter with their providers. Questions about sleep and eating habits were seen as potentially educational for portal users unfamiliar with the non-mood symptoms of anxiety and depression. Users did not specify whether the modality of screening (portal vs. in-person) impacted the potential for increased self-awareness.


*“So I like that way of asking because I think even if you for example, don’t have depression related insomnia, it can help that patients start to think about maybe changes that they are experiencing that do apply”.*
Participant 9 (34-year-old male)


*“A lot of people, they wake up and you have mood swings, you got a moody day. You might go to sleep with an attitude, and you wake up with that attitude. Sometimes, it might be part of depression and you won’t even know that it’s depression”*
Participant 4 (25-year-old female)

#### 3.2.5. Process

The process domain includes constructs that describe how a change should be implemented. The identified constructs included planning and engaging. Innovation participants was a frequently identified subcode.

##### Facilitator: Bundling Anxiety and Depression Screening as Part of a Scheduled Visit

Most patients, especially those who do not already use the patient portal, expressed a strong preference that anxiety and depression screening be bundled with a scheduled clinic visit. Suggestions included pairing portal-based screening with the appointment reminder emails sent out 1–2 days in advance, completing screening in the waiting room via tablet, or including screening as part of the medical assistant intake process. The ability to follow-up results with the care team in-person was the most commonly cited reason for this preference. Patients felt similarly about reminders, with most preferring that screening reminders be sent a few days before their scheduled visits.


*“You wouldn’t want to send an email for the appointment reminder and then a separate email for this, right? I mean, to me it doesn’t feel like a full thought. It seems disconnected like this group’s working over here and that group’s working over there and they don’t talk and they can’t work together so you get two emails, right?”*
Participant 9 (34-year-old male)

##### Facilitator: Communicating an Action Plan for Screening Results

Patients expressed that they would be more likely to participate in portal-based screening if they knew what happens to screening results. Otherwise, there was a perception that the results may add a depression or anxiety diagnosis to their record without being reviewed by a care team member. Knowledge that a trusted member of the care team would eventually receive and act upon the screening results increased interest. Patients also expressed greater interest in screening if they knew that the screening could potentially lead to treatment or referral.


*“My only concern will be that I just want to get the accurate help. I just don’t want to be misdiagnosed through the chart as opposed to talking to someone at the clinic.”*
Participant 1 (52-year-old male)

## 4. Discussion

This qualitative study identified patient-focused facilitators and barriers to the implementation of a portal-based anxiety and depression screening tool within an HIV clinic. Eight themes were identified as facilitators to portal-based screening within four domains (intervention characteristics, inner setting, characteristics of individuals, and process). Five themes were identified as barriers to portal-based screening across the intervention characteristics, outer setting, and inner setting domains. There was an overall positive reception to the design and adaptability of the screening tool itself, although the complexity of navigating the larger electronic health messaging system was a barrier. Respondents expressed doubt about both the patient and provider’s abilities to effectively utilize the electronic messaging system. Further limiting the anticipated uptake, the respondents expressed fears of mental health stigmatization from the healthcare system if privacy and confidentiality could not be maintained. Despite these fears, patients expressed the high compatibility of anxiety and depression screening with routine HIV care. Patients viewed screening as a critical step toward anxiety and depression management, as well as a potential educational opportunity for those unaware of their own depressive symptoms. Patients identified a number of engagement strategies within the process domain, including the bundling of screening with routine clinic visits and the communication of an action plan for screening results that could help to overcome barriers to portal-based screening.

The identified themes are largely congruent with other studies of general anxiety and depression screening practices in HIV and non-HIV specialty care that utilized the CFIR framework. The common barriers across studies that were expressed by patients and staff include (1) a desire for a detailed follow-up protocol to ensure positive screens are met with provider actions [34,35,36] (Process), (2) data storage, privacy, and technical feasibility concerns [34,35,36] (Intervention characteristics), and (3) concerns about the stigmatization of mental health [35,46,47] (Outer Setting). Shared facilitating themes include the general simplicity of anxiety and depression screening tools [46,47] (Intervention characteristics) and the perceived positive value of mental health screening (Characteristics of individuals) [34,36,46].

Unique to respondents in this study was the consistently positive reception to mental health screening as an appropriate and acceptable facet of routine HIV care. Views on the appropriateness and acceptability of mental health screening in other specialty settings are often mixed [34,35,36]. It is not clear whether this is a site-specific finding or reflective of the acceptability of broader mental health screening in HIV care. This finding is consistent with the high acceptability of mental health treatment integration into HIV care [48,49,50,51].

Another intriguing concern raised by participants is the protection of patients experiencing suicidality or other imminent risk, here coded within the inner setting domain. Patients can communicate thoughts of self-harm via patient portals used for communication in mental health programs [52]. Uncertainty about how to manage suicidality may present a major barrier to the implementation of portal-based screening [34].

Based on these themes, we propose a set of strategies to address patient needs during the implementation process:

### 4.1. Proposed Implementation Strategies

#### 4.1.1. Strategy 1: Promote Confidentiality

It should be clear that any anxiety and depression screening results are seen only by the care team. It should be clarified that screening results are routed to a provider for a review and do not automatically link to diagnoses in the medical record. Staff that are involved in processing screening results should be educated about patient confidentiality [53].

#### 4.1.2. Strategy 2: Promote Individual Adaptability

Persistent access to the screening portal outside of planned screening periods or scheduled clinic visits should be provided to demonstrate that the tool will always be available to patients as needed. It should be included in patient messaging that portal-based screening can be performed on a patient’s schedule or that screening can be completed more often if needed [54].

#### 4.1.3. Strategy 3: Emphasize Linkage to Mental Health Resources within Screening Messaging

Message framing can have a significant impact on the uptake of mental health services [55,56]. Information about general anxiety and depression treatment, available embedded mental health services, and available referral-based mental health services should be included early in messaging and should frame screening as a means of linkage to these resources. Patients expressed a greater likelihood of engaging with screening if they were aware of potential treatment options. Informing patients of which staff member will be receiving their results may provide reassurance that the results will be acted upon.

#### 4.1.4. Strategy 4: Bundle Anxiety and Depression Screening with Appointments

Reminders to complete portal-based screening should be coupled with appointment reminders, ideally within the same phone call, text, or email [57]. Responses varied as to the timing and frequency of reminders but a general time frame of 1–3 days before a scheduled appointment appears to be the most favored approach. An option to complete screening in the waiting room and/or with a medical assistant should be provided for those who forget to complete screening or elect not to use the portal. Paper and tablet-based screening in the waiting room both appear acceptable.

#### 4.1.5. Strategy 5: Facilitate and Promote Timely Relay of Positive Screening Results to Providers

An anticipated response time alongside the patient-facing screening results should be provided if possible. At the back end, it should be ensured that positive screening results are relayed to providers in a timely manner. Integrated provider-facing alerts within the EHR may reduce the number of steps needed to process results and reduce the workload of clinic staff. Sending redundant alerts to multiple providers or escalating unopened results to a designated staff member can ensure that positive results are not overlooked. Prior to implementation, staff and provider input on result processing should be sought out to ensure that these workflows are acceptable to all parties, especially a plan for what to do if suicidality is expressed via the portal.

#### 4.1.6. Strategy 6: Minimize Electronic Patient Portal Navigation Steps

By recognizing that patient portal systems may differ widely in structure, it is important to prioritize portal design that reduces the total number of navigation steps for users. This includes minimizing clicks, login pages, and menu navigation [58]. It is also important to provide direct links to the anxiety and depression screening portal within reminder emails and texts, bypassing navigation of the rest of the electronic health messaging system if possible. If performing screening on tablets in the clinic, it is crucial to ensure that the portal is pulled up for the patient. Screening results should appear upon completion of the screening tests and should not require navigation to a new page if possible. Electronic screening within a web browser may be preferable over downloading a separate application for the electronic health messaging system and should reduce technical support needs.

#### 4.1.7. Strategy 7: Provide Technical Support

Satisfaction with portals is improved when patients are directly educated on their use [58,59]. Medical assistants or administrative staff can provide patients with brief education on how to sign up for electronic messaging systems and how to navigate the system. Many hospital systems or electronic health records will have existing patient-facing support systems ranging from online troubleshooting guides to one-on-one assistance. Directing users to these resources can minimize technical support burdens on HIV staff. Links to online resources or phone numbers for virtual support could be included in introductory text or in reminder notifications.

### 4.2. Strengths and Limitations

This qualitative study provides a valuable perspective on electronic mental health screening practices from adults with HIV, including those experiencing depression and anxiety. While the participation of patients with and without anxiety and depression added valuable perspectives from both sides of a mental health diagnosis, no major thematic differences were identified between the two groups. It is unknown if a larger sample would reveal divergent perspectives on screening practices.

Recruitment and scheduling difficulties resulted in interviews with 1 to 3 participants rather than exclusively one-on-one interviews or formal large focus groups. While the research team elected to proceed with group interviews, this may have introduced bias into the participant responses and limited the reproducibility of the results. It is also possible that the financial incentives for participants introduced bias in favor of the intervention.

The single-site nature of this study also limits its generalizability to other clinic settings. The participants were all English-speaking and predominantly black men. Even HIV clinic sites with highly similar patient populations may require alternative implementation strategies if using a different electronic messaging system or if the uptake of electronic messaging is not widespread. The prevalence of portal use among our participants (*n* = 8, 75%) may be higher than other sites. The participants were also younger than the average adult with HIV (mean age 39) and may thus be more comfortable with electronic messaging in general [52]. Differences in the length of time since HIV diagnosis were also not considered during the analysis. This may result in the over-representation of younger adults with relatively recent HIV diagnoses. However, even this younger, high-use group identified a number of technical challenges with the electronic messaging system, and these were factored into the recommended strategies. While the general simplicity of the screening test itself is unlikely to create significant technical barriers for the user, future work should pay greater attention to patient portal usability testing.

Provider and staff perspectives on barriers and facilitators to portal-based mental health screening will also greatly inform implementation strategy choices, but they are not addressed in this study: additional work from this research group focused on providers and staff has been published separately [39]. Clinics are likely to find that implementation strategies that align with both patient and provider views have the greatest impact on screening practices. Further studies including providers and staff may benefit from specifically focusing on the feasibility of the patient-suggested implementation strategies above. Similarly, future qualitative work on portal-based mental health screening should specifically assess the needs of high-risk patients, including those who report self-harm. Once screening programs are fully implemented, future studies will be needed to determine whether or not portal-based screening has a positive effect on diagnosis rates and clinical outcomes.

## 5. Conclusions

This is the first study to examine patient perspectives on portal-based anxiety and depression screening within an HIV care setting. Patients in an HIV clinic viewed the use of a portal-based anxiety and depression screening tool as highly compatible with routine HIV care and as a valuable part of the treatment cascade. Technical difficulties, follow-up concerns, and a fear of stigmatization were commonly perceived as barriers to the effective use of the portal. The consideration of identified facilitators and barriers when choosing implementation strategies will aid the expansion of portal-based mental health screening at other HIV clinic sites. To that end, we propose a set of implementation strategies, adapted from other care settings, that may help close the gaps in anxiety and depression diagnosis among people living with HIV.

## Figures and Tables

**Figure 1 ijerph-21-00692-f001:**
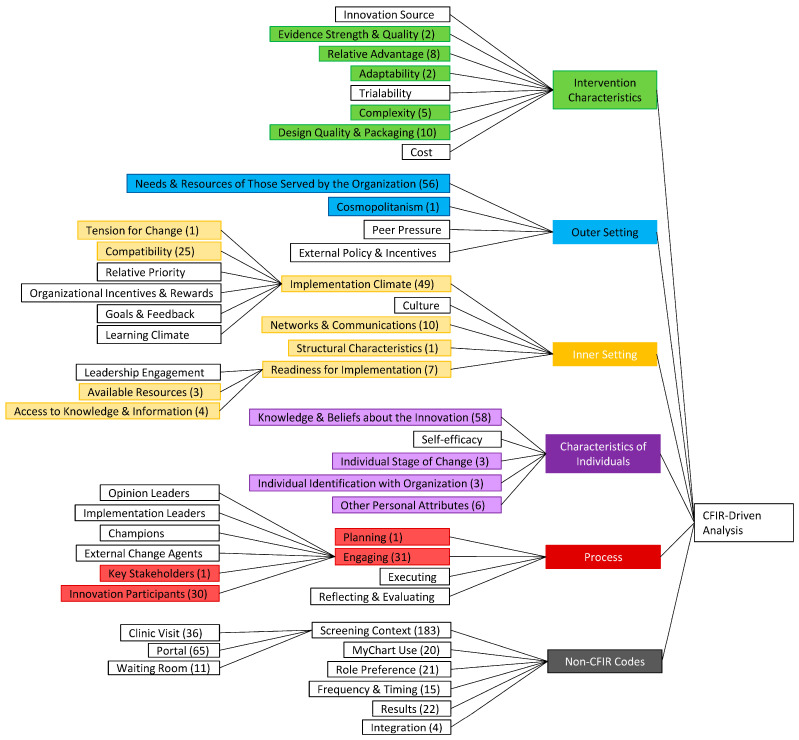
Coding tree of identified CFIR constructs. Identified codes are color-coded by CFIR domain. The number of times the code appeared in interviews is in parentheses.

**Table 1 ijerph-21-00692-t001:** Participant characteristics.

Participant Characteristics (*n* = 12)	*n* (%)
Age (range)	38.8 ± 10.5 (26–57)
Gender identity ^a^	
Male	10 (83.3)
Female	2 (16.7)
Race/Ethnicity ^b^	
Black	9 (75)
Hispanic or Latinx	1 (8.3)
Hawaiian/Pacific Islander	1 (8.3)
White	1 (8.3)
Sexual orientation ^c^	
Gay	6 (50)
Bisexual	3 (25)
Heterosexual/Straight	3 (25)
Prefer not to answer	1 (8.3)
Years living with HIV (range, *n* = 10)	10.6 ± 8.1 (1–22)
Anxiety or depression diagnosis	6 (50)
Employment status	
Employed	8 (66.7)
Unemployed	3 (25)
Retired	1 (8.3)
Marital status	
Single, never married	6 (50)
Single, separated	2 (16.7)
Unmarried, living with a partner	3 (25)
Married	1 (8.3)
Education	
High school graduate/GED	5 (41.7)
Some college or technical school	4 (33.3)
4-year college degree	3 (25)
Electronic portal usage	
Current use	8 (67)
Intent to use in the future	2 (16.7)

^a^ Participants could respond freely; ^b^ Participants could select all races/ethnicities that applied; ^c^ Participants could select all sexual orientation responses that applied.

**Table 2 ijerph-21-00692-t002:** Facilitators and barriers by CFIR domain.

Domain	Facilitators	Barriers
Intervention characteristics	Absence of alternative screening methods Approachable design Perceived adaptability	Difficult-to-navigate patient portal system
Outer setting		Stigmatization from the healthcare system
Inner setting	High compatibility with HIV care	Lack of technical support Utility dependent on care team response time Novel communication method
Characteristics of individuals	Screening viewed as an important part of linkage to treatment Screening may increase self-awareness	
Process	Bundling screening with scheduled visits Communicating an action plan for results	

## Data Availability

Data available from the corresponding author upon reasonable request.

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
