# Peer review of "Patient Perspectives on Portal-Based Anxiety and Depression Screening in HIV Care: A Qualitative Study Using the Consolidated Framework for Implementation Research"

_ijerph, 2024, doi:10.3390/ijerph21060692_

Round 1

Reviewer 1 Report

Comments and Suggestions for Authors

Patient Perspectives on Portal-Based Anxiety and Depression Screening in HIV Care: A Qualitative Study Using the Consolidated Framework for Implementation Research

Thank you for the opportunity to review this relevant and important contribution in HIV care and mental health. Despite this, I believe that a few changes would improve the overall quality of the article:

1.        Introduction section needs further information on why high prevalence of anxiety and depression symptoms exists.

2.        Authors mention that Patient portals are secure websites. How is this security assured?

3.        Objectives need to be further specified. It is not sufficient to say “the aim of this qualitative study is to characterize the patient-perceived facilitators and barriers to portal-based anxiety and depression screening within an HIV care setting using the Consolidated Frame-work for Implementation Research”.

4.        Methodology: authors must clarify and provide information regarding the Consolidated criteria for reporting qualitative research (COREQ), especially in topics such as reflexivity, study design and analysis and findings.

5.        More specific information regarding patient interviews which were conducted by a trained clinical health psychologist (BB) using a predetermined interview protocol, is necessary.

6.        Results: A table summarizing themes and number of occurrences would be beneficial to the reader.

7.        Implications of these results should be better discussed.

Best wishes.

Author Response

Line numbers reflect the most recent draft which will be uploaded alongside responses to reviewers.

REVIEWER 1

  1. Introduction section needs further information on why high prevalence of anxiety and depression symptoms exists.

We have added a sentence explaining that the high prevalence of anxiety and depression has been hypothesized to be due to overlapping sociodemographic vulnerability, medical comorbidities, and HIV-associated stigma (Line 40-42).

  1. Authors mention that Patient portals are secure websites. How is this security assured?

We have removed the word “secure” and now describe patient portals as “encrypted” websites that allow messaging in compliance with the Health Insurance Portability and Accountability Act (Line 59-61)

  1. Objectives need to be further specified. It is not sufficient to say “the aim of this qualitative study is to characterize the patient-perceived facilitators and barriers to portal-based anxiety and depression screening within an HIV care setting using the Consolidated Frame-work for Implementation Research.”

We have provided more details in the objectives, stating that this formative qualitative study was done to inform strategies for implementation of mental health screening (Line 82-87).

  1. Methodology: authors must clarify and provide information regarding the Consolidated criteria for reporting qualitative research (COREQ), especially in topics such as reflexivity, study design and analysis and findings.

Thank you for this suggestion. We have added details throughout the methodology using the suggested COREQ checklist as a guide. A completed COREQ checklist has also been submitted alongside this response.

  1. More specific information regarding patient interviews which were conducted by a trained clinical health psychologist (BB) using a predetermined interview protocol, is necessary. – We appreciate the need for more details and have added more details to the methodology. The protocol was developed iteratively by the study team using the cited CFIR Interview Guide Tool. (Line 134-135). Additional information about the interviewer was also added in line with recommended COREQ criteria. (Line 133-134)

  1. Results: A table summarizing themes and number of occurrences would be beneficial to the reader. We agree and have added a CFIR coding tree as Figure 1 which itemizes the themes observed and number of occurrences. The format of this figure is based on similar models used in the literature (DOI: 10.1186/s43058-021-00150-9 and 1186/s12913-023-09209-w).

  1. Implications of these results should be better discussed.

We have added more details and references to the introduction and discussion contextualizing the intervention and results. Future directions have also been added to the discussion. (Line 522-524) Additional text highlighting the novelty of the study and the need to improve anxiety and depression diagnosis within HIV care has also been added to the conclusion. (Lines 526-527, 533-535)

Reviewer 2 Report

Comments and Suggestions for Authors

Dear authors,

First, I would like to congratulate you for choosing the topic of the study, as results could contribute to understanding the correlation between these factors. I hope you find my comments helpful. The plagiarism report is attached for additional modifications (29%). In detail, my comments are as follows:

Introduction

-        Line 37: Βrackets [1, 2, 4] should be placed after the full stop. Do the same throughout the text.

-        The introduction is too brief and unconvincing to both identify the gap in the literature that this study seeks to fill and to outline and assess the fundamental issue and body of scientific knowledge. Which particular research gap is addressed in this paper?

-        For example, for anxiety and depression there is only a small reference to the respective percentages in the first five lines of the text and nothing more.

-        Lines 39-41: Why “these conditions are often underdiagnosed” and why “the need to screen for and treat anxiety and depression in PWH is great”?

-        The introduction should highlight the study's importance and provide a brief explanation of the study's overall background. When compared to other published content, what does it bring to the topic area? Examining the condition of the field's research in detail and mentioning any significant works is crucial. When necessary, please draw attention to contradictory and controversial findings from scientific studies.

Materials and Methods

-        Line 87: Please add “Informed consent was obtained from all subjects involved in the study.” Written informed consent for publication must be obtained from participating patients who can be identified (including by the patients themselves). Please state “Written informed consent has been obtained from the patient(s) to publish this paper” if applicable.

-        Study sample: The length of time since their diagnosis was not an exclusion criterion; Given that rates of these disorders change over the course of someone with HIV's life. If this factor was not considered at all, it should be mentioned in the study limitations and of course analyzed in the discussion.

-        Lines 125-126: Did the participants know before their participation that they would be paid? perhaps payment as an incentive initially influenced the choice of participants. Reference should be made to this in the discussion.

Discussion

-        The results and how they might be interpreted in light of earlier research and working hypotheses should be reported by the authors. Results and this section should be combined. This discussion mostly consists of generalizations; while it falls short of comparing the results of this study with the body of international literature. For example, the proposed strategies should be supported by literature. for example, the proposed strategies should be supported by literature. That is, the relevant references on which each proposed strategy is based and supported should be added.

-        Future research directions may also be mentioned.

Conclusions

-        Please explain how the evidence and arguments offered support the findings or do not support them. Please specify which specific methods were used to address each of the primary concerns that were raised. What does it add to the subject area compared with other published material?

References

-        Please correct the reference list according to the instructions for authors. For example: Author 1, A.B.; Author 2, C.D. Title of the article. Abbreviated Journal Name YearVolume, page range.

Comments on the Quality of English Language

Minor editing of English language required.

Author Response

Line numbers reflect the most recent draft which will be submitted alongside responses to reviewers

Reviewer 2

Introduction

  1. Line 37: Βrackets [1, 2, 4] should be placed after the full stop. Do the same throughout the text. – All brackets for references have been moved to after the full stop.

  1. The introduction is too brief and unconvincing to both identify the gap in the literature that this study seeks to fill and to outline and assess the fundamental issue and body of scientific knowledge. Which particular research gap is addressed in this paper? – Additional background information on the low detection rates of anxiety and depression in HIV has been added to the introduction in line with requests from other reviewers.(Line 45-46) This helps to clarify the research gap which is now more clearly defined in the last paragraph of the introduction: There is no information in the scientific literature on patient perspectives on portal-based depression and anxiety screening within HIV care.(Lines 78-82)

  1. For example, for anxiety and depression there is only a small reference to the respective percentages in the first five lines of the text and nothing more. – Additional information about etiology, prevalence, and underdiagnosis of depression and anxiety in PWH has been added to the introduction. (Lines 40-42, 45-46)

  1. Lines 39-41: Why “these conditions are often underdiagnosed” and why “the need to screen for and treat anxiety and depression in PWH is great”? - Additional information about etiology, prevalence, and underdiagnosis of depression and anxiety in PWH has been added to the introduction. (Lines 40-42, 45-46)

  1. The introduction should highlight the study's importance and provide a brief explanation of the study's overall background. When compared to other published content, what does it bring to the topic area? Examining the condition of the field's research in detail and mentioning any significant works is crucial. When necessary, please draw attention to contradictory and controversial findings from scientific studies.

We have clarified in the last paragraph of the introduction that there is no information in the scientific literature on patient perspectives toward portal-based depression and anxiety screening in HIV care. Therefore, there was limited context to compare/contrast our findings. (Line 78-83)

Materials and Methods

  1. Line 87: Please add “Informed consent was obtained from all subjects involved in the study.” Written informed consent for publication must be obtained from participating patients who can be identified (including by the patients themselves). Please state “Written informed consent has been obtained from the patient(s) to publish this paper” if applicable. For this study, verbal informed consent was obtained from all participants. The verbal consent script was approved by the University of Chicago Institutional Review Board. This is now stated explicitly in the methods section. (Line 101-103)

  1. Study sample: The length of time since their diagnosis was not an exclusion criterion; Given that rates of these disorders change over the course of someone with HIV's life. If this factor was not considered at all, it should be mentioned in the study limitations and of course analyzed in the discussion. – We have added this analysis limitation to the discussion (Line 506-508)

  1. Lines 125-126: Did the participants know before their participation that they would be paid? perhaps payment as an incentive initially influenced the choice of participants. Reference should be made to this in the discussion. – Participants were made aware that they would be paid during the recruitment process. The amount paid is now included in the methodology. This potential for bias has been added to the limitations in the discussion. (Lines 152-153, 497-498)

Discussion

  1. The results and how they might be interpreted in light of earlier research and working hypotheses should be reported by the authors. Results and this section should be combined. This discussion mostly consists of generalizations; while it falls short of comparing the results of this study with the body of international literature. For example, the proposed strategies should be supported by literature. For example, the proposed strategies should be supported by literature. That is, the relevant references on which each proposed strategy is based and supported should be added. –
    We appreciate the feedback to reorganize our findings and be more specific in our discussion. We have added comparisons of identified themes to those found in other international studies on depression and anxiety screening in paragraph 2 of the discussion. (Lines 406-415) We have also highlighted in paragraph 3 one unique difference in this study from the international literature which is that participants were consistently very positive about anxiety and depression screening (vs. the mixed results seen in other contexts). (Line 416-417) Additional references have been added to each of the proposed strategies, highlighting the use of the propose strategies in similar projects or clinical settings.

  1. Future research directions may also be mentioned. – We have added recommendations for future implementation research to the manuscript. An additional call to verify the clinical outcomes of portal-based anxiety and depression screening in HIV care was added to the discussion. (Line 522-524)

Conclusions

  1. Please explain how the evidence and arguments offered support the findings or do not support them. Please specify which specific methods were used to address each of the primary concerns that were raised. What does it add to the subject area compared with other published material? – We have expanded the conclusion to highlight the novelty of this study and the importance of the identified implementation strategies. (Lines 526-537, 533-535) We hope that this reviewer’s concerns are more adequately addressed with additions to the discussion section above including (1) expanded limitation and strengths, (2) future directions, and (3) supporting references within the proposed implementation strategies.

References

Please correct the reference list according to the instructions for authors. For example: Author 1, A.B.; Author 2, C.D. Title of the article. Abbreviated Journal Name Year, Volume, page range. - All references have been reviewed for style, format, and numbering to align with author instructions on the IJERPH webpage.

Reviewer 3 Report

Comments and Suggestions for Authors

Dear authors,
thank you very much for the opportunity to review your manuscript.

Abstract: The abstract should provide more information on the method (design, approach, participant selection, inclusion and exclusion criteria, data collection technique, analysis process).

Keywords: Properly adjusted to MeSH.

Introduction: Informs correctly, from the general to the specific. On page 1 (line 35) the acronyms HIV (PWH) are used without explanation (HIV is later clarified on page 1, line 43), but not PWH.

Methods: In the design section must specify the methodological orientation (phenomenology...). According to the COREQ criteria, a subsection should be included to describe the characteristics of the research team (interviewers, experience with the methodology, relationship with the informants, etc.); separate this information from the sample study subsection. In the analysis subsection, indicate the methodological approach followed for the analysis process (e.g. Glasser & Strauss).

Must be included the rigor criteria of qualitative research, such as the proposals of Corbin & Guba (credibility, transferability,...)

How participants were chosen (e.g. purposive, convenience, consecutive, snowball)? How many participants were in the study? How many people refused to participate or dropped out, and Reasons? Was the interview guide pilot tested? Were repeat interviews carried out? If yes, how many? Were field notes made during and/or after the interview? Were transcripts returned to participants for comment and/or correction? What software, if applicable, was used to manage the data? Did participants provide feedback on the findings?

Results: The table of sociodemographic characteristics in qualitative studies using means, standard deviation, frequencies and percentages does not provide relevant information. It would be desirable to have a table describing the characteristics of each participant (e.g: participant 1: Male, 39 years old, black, Bisexual, unemployed, married, high school graduate). If this table is implemented, it is possible to link subsequent verbatims (participant 4, 8,...) with their sociodemographic characteristics.

The number of codes, categories of analysis and themes identified should be provided. It would be very interesting to show a figure with the coding tree. A figure clarifying the grouping of categories of analysis into themes (as well as reflecting barriers, domains and their grouping into CFIR domains).

It is possible that table 2 could be used for this purpose (but I think it is not correct to include in this table the proposed strategies, this information, in my opinion, should not constitute a result derived from the analysis of the interviews).

Discussion: You should not quote table 2 in the discussion, as I have described in the previous comment, it is possible that the proposed strategies are not results and if it is more relevant as elements of discussion, it is possible that the proposed strategies are not results and if it is more relevant as elements of discussion. It does not seem appropriate to discuss the strategies in a schematic way; it would be more appropriate to do so in an integrated manner in the discussion.

It is not necessary to include a list of acronyms before the references (they have already been explained in the text of the manuscript).

References: Review the style of the references and the DOI or link to the primary source where appropriate.

Author Response

Line numbers reflect the most recent manuscript draft which will be submitted alongside responses to reviewers. 

REVIEWER 3

  1. Abstract: The abstract should provide more information on the method (design, approach, participant selection, inclusion and exclusion criteria, data collection technique, analysis process). - We have provided more information on the methods in the abstract, adding patient characteristics, study design, data collection technique, and the analysis process. (Lines 20-24)

  1. Introduction: Informs correctly, from the general to the specific. On page 1 (line 35) the acronyms HIV (PWH) are used without explanation (HIV is later clarified on page 1, line 43), but not PWH. – The acronyms used in the introduction have been edited for clarity. The list of acronyms at the end of the manuscript has been removed as requested.

  1. Methods: In the design section must specify the methodological orientation (phenomenology...). According to the COREQ criteria, a subsection should be included to describe the characteristics of the research team (interviewers, experience with the methodology, relationship with the informants, etc.); separate this information from the sample study subsection. In the analysis subsection, indicate the methodological approach followed for the analysis process (e.g. Glasser & Strauss).

We have clarified the following items in the methodology: phenomenological methodologic orientation (Line 97-98), characteristics of the interviewer (Line 133-134), relationship between the research team and subjects (Line 127-129), and the methodologic approach for analysis (Line 97-100). We have elected not to create a new subsection for this information to minimize duplication of information and to avoid conflict with the requests of other reviewers. A completed COREQ checklist has been submitted alongside this response.

  1. Must be included the rigor criteria of qualitative research, such as the proposals of Corbin & Guba (credibility, transferability,...) – Additional details have been added throughout the methodology to aid in critical review using the suggested COREQ checklist as a guide. A completed COREQ checklist has been submitted alongside this response.

  1. How participants were chosen (e.g. purposive, convenience, consecutive, snowball)? – We have clarified that a purposive sampling method was used in the methods section. (Line 115-117)

  1. How many participants were in the study? How many people refused to participate or dropped out, and Reasons? – 59 people were approached about the study. 47 people declined to participate. No participants dropped out of the study. Reasons for non-participation were not recorded. This information has been added to the results section on participant characteristics. (Line 174-176)

  1. Was the interview guide pilot tested? – The interview guide was not pilot tested. This has been added to the methods section alongside other recommended clarifications to protocol development. (Line 139)

  1. Were repeat interviews carried out? If yes, how many? – Repeat interviews were not conducted. This clarification has been added to the methods section. (Line 147)

  1. Were field notes made during and/or after the interview? – Additional written records were not maintained to reduce the risk of a privacy breach. This clarification has been added to the methods section. (Line 149-150)

  1. Were transcripts returned to participants for comment and/or correction? – No, transcripts were not provided to participants. This clarification has been added to the methods section. (Line 150-151)

  1. What software, if applicable, was used to manage the data? – Transcript storage and analysis was conducted using Dedoose. This is included in the methods section. (Line 167-169)

  1. Did participants provide feedback on the findings? – Participants were not asked to provide feedback on results. This clarification has been added to the methods section. (Line 150-152)

  1. Results: The table of sociodemographic characteristics in qualitative studies using means, standard deviation, frequencies and percentages does not provide relevant information. It would be desirable to have a table describing the characteristics of each participant (e.g: participant 1: Male, 39 years old, black, Bisexual, unemployed, married, high school graduate). If this table is implemented, it is possible to link subsequent verbatims (participant 4, 8,...) with their sociodemographic characteristics. –

The authors have chosen not to include a detailed table of participant information in order to protect participant privacy. Participants were recruited from a single clinic that largely serves patients from a small geographic area. This increases the risk that participants could be identified from only a few characteristics. However, we agree that more information about respondents is warranted and have added participant sex and age to the representative quotes included throughout the results section.

  1. The number of codes, categories of analysis and themes identified should be provided. It would be very interesting to show a figure with the coding tree. A figure clarifying the grouping of categories of analysis into themes (as well as reflecting barriers, domains and their grouping into CFIR domains). - In response to this comment and those from other reviewers, a CFIR coding tree has been added as Figure 1 to quantify the themes observed and number of occurrences. Format of this figure is based on similar models used in the literature (DOI: 10.1186/s43058-021-00150-9 and 1186/s12913-023-09209-w)

  1. It is possible that table 2 could be used for this purpose (but I think it is not correct to include in this table the proposed strategies, this information, in my opinion, should not constitute a result derived from the analysis of the interviews).

Table 2 has been revised and re-titled to only include the facilitators and barriers identified by CFIR domain. The proposed implementation strategies have been relegated to the discussion as suggested.

  1. Discussion: You should not quote table 2 in the discussion, as I have described in the previous comment, it is possible that the proposed strategies are not results and if it is more relevant as elements of discussion, it is possible that the proposed strategies are not results and if it is more relevant as elements of discussion. It does not seem appropriate to discuss the strategies in a schematic way; it would be more appropriate to do so in an integrated manner in the discussion. –

The reference to Table 2 in the discussion has been removed to more clearly separate the proposed implementation strategies in the discussion section from the results. The proposed implementation strategies remain integrated in the text of the discussion as suggested.

  1. It is not necessary to include a list of acronyms before the references (they have already been explained in the text of the manuscript). – The list of acronyms has been removed as requested.

References: Review the style of the references and the DOI or link to the primary source where appropriate. – All references have been reviewed for style, format, and numbering to align with author instructions on the IJERPH webpage.

Round 2

Reviewer 1 Report

Comments and Suggestions for Authors

Thank you for implementing all the requested changes. I believe the article is now fit for publication.

Best wishes.

Author Response

The entire study team thanks you for your thoughtful revisions. The manuscript is certainly stronger as a result. 

Reviewer 3 Report

Comments and Suggestions for Authors

Dear authors,

thank you very much for the improvements in the manuscript.

Abstract: The aim should be included in the abstract using the same terms as in the text of the manuscript.

Results: Figure 1 cited in the text has not been included. However, the potential location should be as close as possible to the text in which it is cited.  Table 2 should also be placed as close as possible after it is cited in the text. However, in this revision of the manuscript the citation does not appear in the text (it is crossed out). Table 2 must correspond to results and be cited in the results section (before detailed results with the verbatims). The organization of these results following the proposed domains would be improved by integrating each facilitator and each barrier with a single subheading in each domain (e.g. Domain 1 Intervention characteristics - Facilitartors: 1. Absence of alternative anxiety and depression screening methods ; 2 Simple and approachable portal design...). (only once the heading Facilitators). Same for Barriers).

In Table 2 it is not necessary to describe barriers and facilitators in each row (the column heading identifies this aspect).

Author Response

Reviewer 3

Abstract: The aim should be included in the abstract using the same terms as in the text of the manuscript. – We have updated the description of our research aim in the abstract to mirror the language used in the introduction.

Results: Figure 1 cited in the text has not been included. However, the potential location should be as close as possible to the text in which it is cited.  – The suggested location of Figure 1 has been moved to be close to the place in the text where Figure 1 is cited. Figure 1 itself is uploaded to the MDPI system for review, we apologize if it was not viewable for reviewers. A copy of Figure 1 has been attached to this response.

Table 2 should also be placed as close as possible after it is cited in the text. However, in this revision of the manuscript the citation does not appear in the text (it is crossed out). Table 2 must correspond to results and be cited in the results section (before detailed results with the verbatims). – Thank you for catching this mistake. A reference to Table 2 has been added back to the text early in the results section before the detailed results. The location of Table 2 has also been moved earlier in the results section to correspond to the citation in the text.

The organization of these results following the proposed domains would be improved by integrating each facilitator and each barrier with a single subheading in each domain (e.g. Domain 1 Intervention characteristics - Facilitators: 1. Absence of alternative anxiety and depression screening methods ; 2 Simple and approachable portal design...). (only once the heading Facilitators). Same for Barriers). In Table 2 it is not necessary to describe barriers and facilitators in each row (the column heading identifies this aspect). - The content of Table 2 has been simplified by removing the subheadings as suggested
